# Phylogenetic Analysis of Varicella–Zoster Virus in Cerebrospinal Fluid from Individuals with Acute Central Nervous System Infection: An Exploratory Study

**DOI:** 10.3390/v17020286

**Published:** 2025-02-19

**Authors:** Heuder G. O. Paião, Antônio C. da Costa, Noely E. Ferreira, Layla Honorato, Bianca M. dos Santos, Maria L. M. de Matos, Renan B. Domingues, Carlos A. Senne, Amanda de O. Lopes, Vanessa S. de Paula, Steven S. Witkin, Tânia R. Tozetto-Mendoza, Maria Cássia Mendes-Correa

**Affiliations:** 1Laboratório de Virologia (LIM52), Instituto de Medicina Tropical de São Paulo, Faculdade de Medicina da Universidade de São Paulo, Av. Dr Enéas de Carvalho Aguiar, 470, São Paulo 05403-000, Brazil; charlysbr@yahoo.com.br (A.C.d.C.); noely.evangelista@hc.fm.usp.br (N.E.F.); layla.honorato@usp.br (L.H.); switkin@med.cornell.edu (S.S.W.); tozetto@usp.br (T.R.T.-M.); maria.cassia@hc.fm.usp.br (M.C.M.-C.); 2Departamento de Infectologia e Medicina Tropical, Faculdade de Medicina da Universidade de São Paulo, Av. Dr Arnaldo, 455, São Paulo 01246-903, Brazil; 3Serviço de Cuidados Paliativos do Hospital das Clínicas da Faculdade de Medicina da Universidade de São Paulo, R. Cotoxó, 1142, São Paulo 05021-001, Brazil; bia.martinss13@gmail.com; 4Faculdade de Medicina, Universidade de São Caetano do Sul, R. Santo Antônio, 50, São Caetano do Sul 09521-160, Brazil; maria.matos318@hc.fm.usp.br; 5Laboratório Senne Liquor, Av. Angélica, 2071, São Paulo 01239-030, Brazil; renan.domingues@senneliquor.com.br (R.B.D.); carlos.senne@senneliquor.com.br (C.A.S.); 6Laboratório de Virologia Molecular do Instituto Oswaldo Cruz, Av. Brasil, 4365, Rio de Janeiro 21040-900, Brazil; amanda.lopes.fiocruz@gmail.com (A.d.O.L.); vdepaula@ioc.fiocruz.br (V.S.d.P.); 7Department of Obstetrics and Gynecology, Weill Cornell Medicine, New York, NY 10001, USA

**Keywords:** Varicella–Zoster virus, central nervous system, Clade variation

## Abstract

Background: There is scarce information on Varicella–Zoster virus genetic variability in individuals with acute central nervous system infection in Brazil. The objective of this study was the molecular characterization of Varicella–Zoster virus isolates in cerebrospinal fluid from individuals with acute central nervous system infection. Methods: Cerebrospinal fluid samples were collected from individuals evaluated in emergency and community healthcare services in São Paulo, Brazil. Varicella–Zoster virus identification was performed using commercial platforms Biofire-FilmArray Meningitis/Encephalitis (BioMérieux, Craponne, France) and XGEN-UMLTI-N9^®^ (Mobius Life, Pinhais, Brazil). Positive samples were further characterized as wild-type or vaccine-strain by a real-time polymerase chain reaction assay that targeted a single nucleotide polymorphism in open reading frame 62. We also estimated the mean genetic distance and phylogenetic reconstruction based on open reading frames 22, 38, 54, and 62 in relation to sequences of intercontinentally circulating Varicella–Zoster virus isolates. Results: Among the 600 cerebrospinal fluid samples, we identified Varicella–Zoster virus in 30 (5%) samples. None were positive for the vaccine-strain. Twelve samples were sequenced and phylogenetically classified into Clades 1 (41.7%), 2 (25%), 3 (8.3%), 5 (16.7%), or 6 (8%). Conclusion: Enhanced characterization of circulating Varicella–Zoster virus Clades in Brazil identified previously unreported Clades 2 and 6 as well as three other Clades disseminated intercontinentally. These findings reinforce the importance of Varicella–Zoster virus molecular surveillance in cerebrospinal fluid.

## 1. Introduction

Varicella–Zoster virus (VZV), also called human herpes virus 3 (HHV-3), is an enveloped DNA virus that belongs to the *Alphaherpesvirus* family. It primarily causes varicella (chickenpox), usually a self-limiting condition [1]. However, after infection resolution, VZV can remain latent in cells of the dorsal root ganglia and reactivate under immunosuppression conditions [2]. Its reactivation can result in herpes zoster, also known as shingles, usually in adulthood. VZV infection or reactivation also can lead to neurological complications of varying severity, in both varicella and herpes zoster presentations [3,4]. VZV is perceived as one of the most frequent viral agents associated with acute central nervous system infection (ACNI) [5]. However, its presence in this condition often goes undiagnosed [6,7,8].

Phylogenetic analysis of VZV isolates in different biological specimens has resulted in their discrimination into nine Clades worldwide (referred to as Clades 1 to 9) [9,10,11]. Data on VZV molecular characterization are limited or unexplored in many regions of the world, including Brazil, where only Clades 1, 3, and 5 have been reported [12,13]. The global distribution of various VZV Clades suggests possible differences in its worldwide dispersion [14,15]. In addition, the potential risk associated with the administration of the VZV live attenuated Oka vaccine (v-Oka) strain remains incompletely evaluated. Although the risk of disease from the v-Oka vaccine is considered low and this strain has not been described in Brazil [12,16], the limited availability of genetic characterization studies on circulating VZV isolates hampers the accurate assessment of its true impact in many countries [17,18].

The genetic characterization of VZV has been based on Clade analyses of specific DNA sequences from open reading frames (ORFs) in the VZV genome, typically involving the analysis of three or more ORFs. Most studies focus on the analysis of ORF 22, ORF 38, ORF 54, and ORF 62 to trace the phylogenetic proximity of circulating VZV viral isolates [11,19,20]. These ORFs encode biologically significant proteins: ORF 22 for a large tegument protein, ORF 38 for a tegument protein, ORF 54 for a capsid portal protein, and ORF 62 for a transcriptional regulator protein [21]. Despite the limited number of reports on mutations and recombination events among VZV strains, these selected ORFs are considered sufficiently polymorphic for phylogenetic analysis [22,23].

The present study describes the profile of VZV isolates identified in cerebrospinal fluid (CSF) from Brazilian individuals presenting with ACNI.

## 2. Materials and Methods

### 2.1. Study Design and Setting

This was a cross-sectional study, which included retrospectively analyzed CSF samples from 600 patients with clinical suspicion of acute infectious encephalitis or meningitis in Sao Paulo, Brazil. Samples were collected within 2–5 days from the onset of symptoms, between 2018 and 2019. All subjects were seen at emergency and community healthcare services in the city of São Paulo. All CSF samples underwent a complete cell count, differential leucocyte count, and biochemistry tests (including glucose, protein, and lactate concentrations) to analyze CSF parameters. For pathogen analysis, the following tests were performed: the examination of Gram-stained and India Ink smears; bacterial and fungal cultures; and the use of the BioFire^®^ FilmArray Meningitis/Encephalitis Panel (BioMérieux, Craponne, France) and XGEN Multiplex Neuro 9 (Mobius Life, Pinhais, Brazil), as described elsewhere [24].

Clinical and laboratory information was obtained from medical records and attached to the REDCap (Research Electronic Data Capture) 14.5.21v platform database. All the CSF samples were stored at −80 °C in the Laboratory of Virology (LIM/52) of the Instituto de Medicina Tropical, Faculdade de Medicina da Universidade de São Paulo (IMT-FMUSP) for subsequent molecular analysis.

The workflow summary is presented in Figure 1.

### 2.2. VZV Identification

VZV infection was diagnosed by using the qualitative commercial platforms Biofire FilmArray Meningitis/Encephalitis Panel (BioMérieux, Craponne, France) and XGEN Multiplex Neuro 9 (Mobius Life, Pinhais, Brazil). The FilmArray Meningitis/Encephalitis Panel (Part Number RFIT-ASY-0118) is a multiplex in vitro diagnostic test for the simultaneous detection of 14 pathogens directly from CSF specimens. These pathogens include *Escherichia coli K1*, *Haemophilus influenzae*, *Listeria monocytogenes*, *Neisseria meningitidis*, *Streptococcus pneumoniae*, *Streptococcus agalactiae*, *Cryptococcus neoformans*/*Cryptococcus gattii*, cytomegalovirus (CMV), Enterovirus (EV; A–D species), herpes simplex virus 1 and 2 (HSV 1 and 2), human herpesvirus 6 (HHV-6), human parechovirus (HPeV), and VZV. Similarly, the XGEN Multiplex Neuro 9 (Part Number XG-N9-MB) is proficient in simultaneously detecting 11 pathogens, including Human Adenovirus (HAdV), CMV, Epstein–Barr virus (EBV), HSV1 and 2, VZV, HPeV, Parvovirus B19, human herpesvirus 6 and 7 (HHV6 and HHV7), and EV (A–D species).

### 2.3. DNA Extraction and Purification

DNA was extracted and purified from 500 µL of each CSF sample using an automated system extractor (NUCLISENS^®^ EASYMAG^®^ bioMérieux, Craponne, France), according to the manufacturer’s instructions. The viability of DNA amplification was assessed by using an internal control, as previously described [25]. DNA samples positive for VZV were identified numerically (for example, VZV001).

### 2.4. Discrimination Test for v-Oka Vaccine Strain

A real-time polymerase chain reaction (PCR) test was performed using primers to amplify a 62 bp product encompassing a single nucleotide polymorphism (SNP) at position 107252 (forward primer, VZ62TF, 5′-ACT GGA GCC CGT TGC CTC-3′; reverse primer, VZ62TR, 5′-TCC TAC AGA GTC TCC GCA GAG C-3′). Two fluorogenic MGB probes were designed with different fluorescent dyes to allow single-tube genotyping. One probe was targeted to the wild-type strains (WT-VZ62T, 5′-6FAM-TTG CCA GCA TGG C-MGB-3′), and one was targeted to v-Oka strains (O-VZ62T, 5′-VIC-TTG CCG GCA TGG C-MGB), as previously described [26]. The data were analyzed using QuantStudio Design & Analysis Software v.1.4.1.

### 2.5. VZV Genotyping

A semi-nested PCR was performed by using Taq Platinum (Invitrogen Life Technologies, Carlsbad, CA, USA) to obtain the DNA fragments of four ORFs of the VZV genome (ORF 22, ORF 38, ORF 54, and ORF 62) with primers and cycling parameters, as previously described [11,19,20,27,28].

PCR products were purified using a QIAquick PCR Purification Kit (QIAGEN, Hilden, Germany) and semi-quantified with a low DNA mass ladder (Gibco, Waltham, MA, USA). Approximately 20ng of purified PCR were sequenced with a Big Dye™ Terminator v 3.1 Cycle Sequencing Ready Reaction kit (Applied Biosystems, Foster City, CA, USA), according to the manufacturer’s instructions, and using the inner primers of the nested PCR described in Table 1. The Sanger sequencer used was the ABI PRISM^®^ 3500 Genetic Analyzer (Applied Biosystems^®^, Foster City, CA, USA).

The quality of read sequences was assessed in CodonCode Aligner V.10.0.2 (CodonCode Corporation, Dedham, MA, USA).

One hundred and twenty worldwide-representative VZV DNA sequences were obtained from GenBank http://www.ncbi.nlm.nih.gov/ (accessed on 23 June 2024) and used as references for analysis. DNA sequences were aligned by using MAFFT, incorporated into the UGENE 11.01 software [29]. Phylogenetic reconstruction was performed under ML by using W-IQ-TREE [30] with a general time-reversible GTR + G model of rate heterogeneity and 1000 bootstrap replicates.

DNA sequences were analyzed and the VZV Clades were accurately confirmed according to the E-value and the identity of pairs at the National Center for Biotechnology Information (NCBI) by the Basic Local Alignment Search Tool (BLASTn). DNA sequences of VZV isolates obtained in this study were deposited in the GenBank database (NCBI).

### 2.6. Intercontinental Dissemination of VZV Clades

The mean genetic distances (MGDs) among sequences of VZV DNA isolated from different geographic regions of the world and from Brazil, including the ones described in the present study, were analyzed to infer the intercontinental dissemination of VZV Clades. Three hundred and sixty one complete or partial prototypic sequences of the VZV genome deposited in the GenBank were analyzed. The mean genetic distances between the groups were estimated using MEGA version 11 [31] with the Kimura two-parameter model, as described elsewhere [32]. The graphic visualization program was Adobe Illustrator v. CC2022, used to construct the geographic map of the intercontinental circulation of the VZV Clades.

## 3. Results

### 3.1. Participants and Genetic Classification of VZV Isolates

A summary of the results is presented in Figure 2.

#### 3.1.1. VZV Identification

Among the 600 CSFs previously tested, VZV was identified in 30 (5%) samples. There was an equal gender distribution, and the median age was 40 years for females (range: 17 to 72 years) and 41.5 years for males (range: 17 to 79 years). Data on the general characteristics of these patients and the CSF Identification Code are presented in Table 2.

#### 3.1.2. Identification of Other Viruses

Along with the identification of VZV, other microorganisms were concomitantly identified by using the commercial platforms Biofire FilmArray Meningitis/Encephalitis Panel (BioMérieux, Craponne, France) and XGEN Multiplex Neuro 9 (Mobius Life, Pinhais, Brazil). Co-infection was observed in three cases: two with Epstein–Barr virus (samples VZV006 and VZV012), and one with Human Adenovirus (HAdv) (sample VZV004) (Table 2).

#### 3.1.3. DNA Extraction Quality Assessment

All extracted VZV DNA samples were suitable for further amplification since 100% of samples were successfully amplified for internal control.

#### 3.1.4. Discrimination Test for v-Oka Vaccine Strain

The VZV strain discrimination test was successful in 25 of the 30 VZV positive samples tested. All tested positive for the wild-type strain. In five samples (VZV002, VZV003, VZV005, VZV010, and VZV011) VZV DNA amplification was not successful, probably due to low viral load. These individuals were already undergoing antiviral therapy (acyclovir) at the time their sample was collected.

#### 3.1.5. VZV Genotyping

Twelve individuals had VZV DNA sequences that were successfully amplified for ORFs 22, 38, 54, and/or ORF 62, enabling VZV Clade classification. The DNA electrophoresis pattern is shown in the Appendix A. The yield of successfully obtained VZV DNA sequences for ORFs 22, 38, 54, and ORF 62 was 50%, 83.8%, 100%, and 50%, respectively. Characteristics of VZV isolates from this study in relation to their genetic classification (Clade), year of CSF collection, and respective GenBank accession number are presented in Table 3. ORFs 22, 38, 54, and/or ORF 62 were not successfully amplified in samples from 18 individuals, likely due to a low viral load, probably because these individuals were already undergoing antiviral therapy at the time their sample was collected, as mentioned above.

### 3.2. Phylogenetic Classification and Nucleotide Identity

Concatenated VZV DNA sequences were phylogenetically classified according to their distance to sequences obtained from GenBank, as follows: Clade 1 (VZV001, VZV015, VZV020, VZV023 and VZV026), Clade 2 (VZV004 and VZV016), Clade 3 (VZV021), Clade 5 (VZV012, VZV027, and VZV030), and Clade 6 (VZV018), in proportions of 41.7%, 16.7%, 8.3%, 25.0%, and 8.3%, respectively. Figure 3 represents the final phylogenetic reconstruction of VZV isolates in this study.

### 3.3. Intercontinental Dissemination of VZV Isolates from Brazil and from Other Origins

The lowest MGDs were observed in sequences from Asia in relation to the available prototypes: Iran, Russia, and South Korea in Clade 1 or 3; India, Iran, and Laos in Clade 2; and Germany, Iran, and Laos in Clade 5. Since few sequences from Clade 6 were available, our sequences showed no divergence in the MGD results. All genetic distances are summarized in Appendix A. The visualization for VZV dispersion is represented in Figure 4.

Other than the stability of the VZV genome, demonstrated in molecular analysis, additional data that may further enhance the consistency of our findings are provided in the Appendix A.

## 4. Discussion

The limited availability of genetic characterization studies on circulating VZV isolates hinders the accurate assessment of their true impact on vulnerable populations, such as patients with ACNI, particularly in healthcare settings in developing countries. To address this limitation, the present exploratory study reports the genetic analysis of VZV isolates in CSF from patients with ACNI seen at different healthcare units in São Paulo, Brazil.

CSF analysis is a classical tool for diagnosing infections of the central nervous system [34,35]. However, a significant challenge in molecular analysis is typically the low viral load in CSF [24]. In the present study, we successfully performed the genetic classification of 12 VZV DNA isolates from CSF samples from among 30 diagnosed cases of VZV. These analyses contributed to the characterization of VZV Clades in CSF samples in Brazil. The vaccine VSV strain, v-Oka, was not detected in any of the samples analyzed. We detected, for the first time in Brazil, Clades 2 and 6, in addition to the identification of three other Clades that exhibit intercontinental dissemination (Clades 1, 3, and 5) [14,15]. We also reported on the co-infection of VZV Clade 2 with EBV and VZV Clade 6 with HAdV. It is necessary to be cautious when interpreting these findings since all these viruses can establish latency and their detection may just be an incidental finding [36,37]. EBV-VZV co-infection in CSF has previously been reported [6,38] but HAdv co-detection with VZV in CSF, to our knowledge, has not been previously published.

In the present study, we identified 30 VZV cases among 600 samples from individuals with ACNI (5%), making it the second most frequent agent identified after Enterovirus, as was previously shown in another study from our group [24]. However, the frequency of confirmed VZV diagnosis in CNS infections in Brazil has varied widely, ranging from 1% to 20% [6,7,39]. A crucial aspect that must be noted is that the actual number of VZV infections in Brazil is likely underestimated, since only severe cases of VZV are mandatorily reported, according to current guidelines from the Brazilian Ministry of Health [40].

The v-Oka immunization program began in North America, Europe, and Asia in the 1990s. In Brazil, this vaccination program was initiated eleven years ago [41]. Vaccination for VZV is associated with a reduction in the number of severe varicella cases reported in many countries [42], as well as a decrease in the number of severe varicella cases among children in the 0 to 5-years-old age group [43]. Reactivation of the live attenuated vaccine v-Oka strain was undetected in all of our analyzed CSF samples.

The dissemination of VZV Clades may provide insights into the latent nature of the virus and the risk of viral reactivation [18,44], particularly in the population over 60 years old. Additionally, an increased number of molecular studies on VZV Clade distribution, along with a further emphasis on the tracking of migratory events, will contribute to the identification of the emergence of new Clades in different regions [15,45].

In the present study, phylogenetic analysis of VZV isolates revealed the presence of Clade 1, Clade 3, and Clade 5, as previously reported by other Brazilian and Latin American studies [12,13,46,47,48]. In addition, Clade 2 was, for the first time, reported in Latin America, as well as Clade 6 in Brazil. Clade 2 had already been reported in the USA [9], Asia [49], Europe [16], and Oceania [11]. Clade 6 had already been reported in Peru [47], Mexico [48], the USA [9], and Europe [43]. Although the global distribution of VZV is well defined [15,45], its genetic diversity has been insufficiently investigated to clarify its differential clinical impact and its potential epidemiological implications.

VZV Clades may exhibit geographic distinctions [50], with certain Clades showing a pronounced predominance in specific regions. Clades 1, 3, and 5 are more prevalent in America [9], Africa [10], Europe [51], and Oceania [11], while Clade 2 is more commonly found in Asia [49,52]. Clade 6 has been described in only a few instances, as it was newly proposed after 2008 [53]. The previously undetected circulation of Clades 2 and 6 may be attributed to limited sampling in earlier studies—or even the absence of studies—rather than a recent introduction of these Clades in Brazil.

VZV interclade genetic variation is less than 0.06%, but still, we performed an analysis to describe the dissemination of circulating VZV Clades in Brazil. Clades were primarily ranked according to their origin in different geographical areas: Europe, Eurasia, Asia, and North America. During this analysis, we became aware of the fact that African VZV prototype sequences are a small portion of all available sequences and that this could represent a bias in our results since VZV is present in all continents [15,54]. In the African continent, few studies have identified Clade 5 [10,38]. Regarding VZV evolution, previous studies demonstrated that VZV and other human herpesviruses co-evolved with humankind. They are thought to have originated in Africa and then spread to Asia and Europe [55], but recent investigations point to a possible European origin of VZV [50,56].

The restricted viral load of VZV in CSF was one of the limiting factors for the number of VZV isolates that could be genetically classified in the present study. Despite this limitation, the protocol used achieved varying success rates, ranging from 50% to 100%, according to the analysis of the different ORFs of the viral genome.

It is important to acknowledge other limitations of the present study. Our study relied on a convenience-based and restricted sample size, constraining the generalization of these findings. In some cases, the viral origin or sequence of the VZV genome was not identified due to the inherently low viral load in the CSF. Furthermore, a review of medical records revealed that some samples sent to our laboratory were from individuals already on anti-viral therapy (typically Acyclovir) prior to sample collection. Regardless of the viral load in the CSF, an unexamined hypothesis in our study to explain the lack of success in sequencing VZV DNA in some samples is that genetic variation in VZV isolates might hinder primer binding to these viral genomes present in the CSF. This may potentially lead to an underestimation of the discovery of new VZV Clades. Lastly, we were not able to ascertain if any of the VZV-positive patients had been undergoing immunosuppressive treatment, which might have exacerbated the expression of latent viruses.

## 5. Conclusions

In summary, our study advances knowledge of the circulating VZV Clade profile in Brazil. Previously unidentified Clades, such as Clades 2 and 6, and three other Clades, which have been previously described intercontinentally, were sequenced and characterized. The present findings reinforce our belief that VZV surveillance must be maintained to establish epidemiological tracking of VZV-associated ACNI and to investigate whether different Clades are associated with specific clinical manifestations of VZV infection.

## Figures and Tables

**Figure 1 viruses-17-00286-f001:**
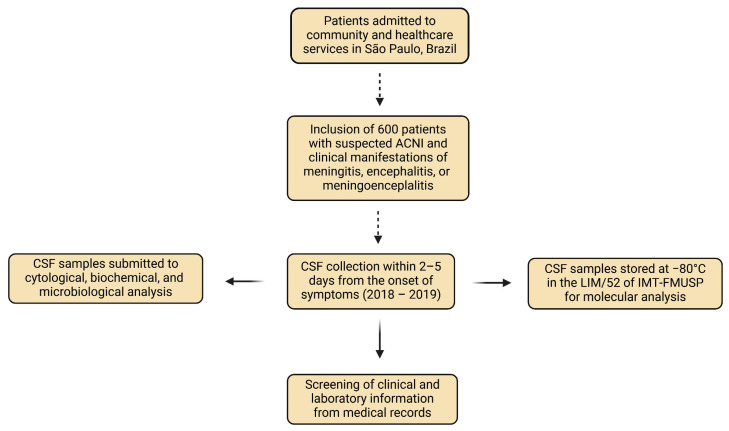
Workflow summary of sample collection and analysis. Created in https://BioRender.com (accessed on 5 February 2025).

**Figure 2 viruses-17-00286-f002:**
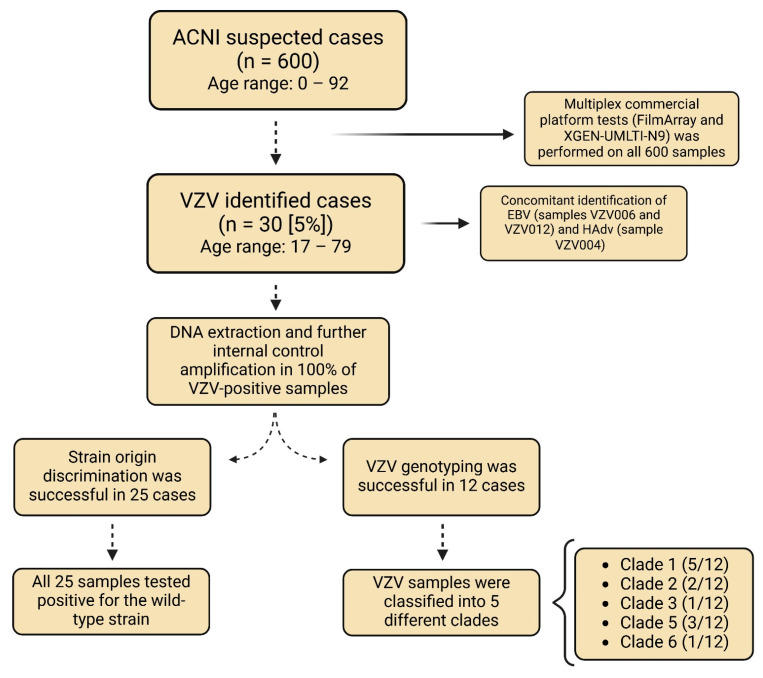
Summary of microorganism identification and genetic classification of VZV isolates. Created in https://BioRender.com (accessed on 5 February 2025).

**Figure 3 viruses-17-00286-f003:**
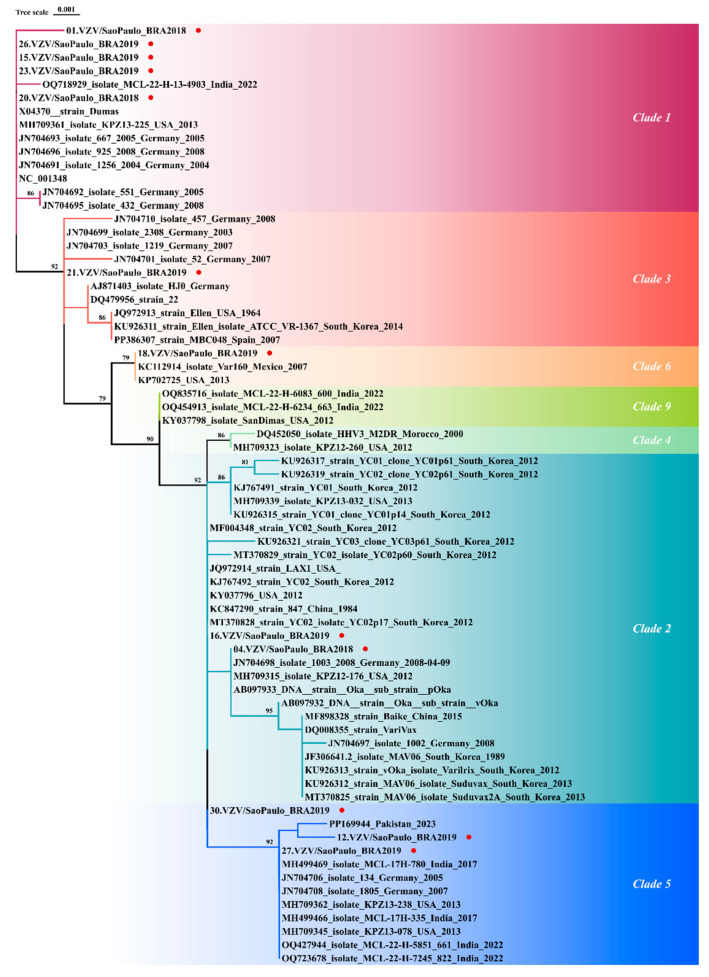
A phylogenetic tree for Varicella–Zoster Virus (VZV) constructed with the W-IQ-Tree, using the maximum likelihood (ML) method, with the GTR+G substitution model. Bootstrap was performed with 1000 replicates. Red circles denote the VZV sequences of this study. GenBank accession numbers for the reference sequences are as follows: AB097932, AB097933, AJ871403, AY548170, DQ008354, DQ008355, DQ452050, DQ479953, DQ479956, DQ479958, DQ479961, DQ479962, DQ674250, EU154348, JF306641, JN704691–JN704710, JQ972913, JQ972914, KC112914, KC847290, KJ767491, KJ767492, KJ808816, KP702725, KU926311–KU926322, KY037796, KY037798, MF004348, MF898328, MH499466–MH499469, MH709311–MH709377, MT370825–MT370830, MW545806–MW545808, OQ427944, OQ454913, OQ718929, OQ723678, OQ835716, OQ871571, OQ916050, PP169944, PP386307, and X04370. GenBank accession numbers for the sequences generated in this study are provided in Table 3.

**Figure 4 viruses-17-00286-f004:**
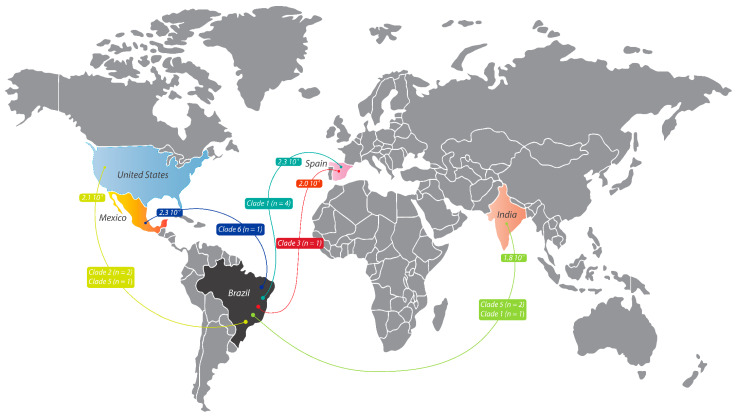
Inferred intercontinental dissemination of VZV isolates. The represented distances are the lowest mean genetic distance (MGD) considering each Clade in relation to the 361 VZV prototype sequences of the GenBank and the 12 VZV sequences from the present study. The mean genetic distances between groups were estimated using MEGA version 11 with the Kimura two-parameter model. 
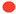
 The red lines represent sequences from Clade 3, 
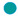
 the turquoise lines represent sequences from Clade 1, 
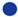
 the blue lines represent sequences from Clade 6, and 
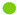
 the green lines represent sequences from Clades 1, 2, and 5. GenBank accession numbers for the reference sequences are as follows: AJ871403, AY379115, AY379116, DQ457052, DQ674250, EU154348, FJ425229–FJ425272, FJ425272, JF306641, JN704690–JN704710, JQ972913, JQ972914, KC112914, KC847290, KF130961, KF130961- KF130965, KF811485, KJ767491, KJ767492, KJ808816, KP702725, KP902647–KP902660, KT360938–KT360942, KU926311–KU926322, KX352173–KX352185, KY037796, KY037797, LT984514–LT984535, MF004348, MF503712–MF503738, MF898328, MH499466–MH499469, MH709309–MH709377, MT358898–MT358918, MT370825–MT370830, MW545806–MW545808, MZ465781–MZ465905, OQ427944, OQ718929, OQ723678, OQ871571, OQ916049, OQ916050, OR689713, OR770642, OR885927, OR898439–OR898444, OR915870, OR958827–OR958829, OR988144–OR988146, OR992036–OR992041, PP003316–PP003321, PP169944, PP261331, PP261332, and PP386307. Adobe Illustrator version CC2022. The origin-colored points do not represent a specific location and are only representative of the country of origin.

**Table 1 viruses-17-00286-t001:** Primers used in semi-nested polymerase chain reaction for Sanger sequencing step.

	Open Reading Frame (ORF)	Sequence 5′-3′	Authors
p22R1f	22	GGGTTTTGTATGAGCGTTGG	Loparev [11] Sauerbrei [27]
p22R1r		CCCCCGAGGTTCGTAATATC	
PstA38	38	TTGAACAATCACGAACCGTT	LaRussa [28]
PstB38		CGGGTGAACCGTATTCTGAG	
Fok54	54	TCCCTTCATGCCCGTTACAT	LaRussa [19]
Nla54		GGAACCCCTGCACCATTAAA	
62F	62	GGCCTTGGAAACCACATGATCG	Luan [20]
62r		CGTCTCCCGTTCCGCATGTAG	

**Table 2 viruses-17-00286-t002:** Characteristics of VZV-positive patients: CSF identification code, sex, age, presence of co-infection, and general molecular data.

Sample	Sex	Age	Co-Infection	Origin Type	Clade	Total bp Size (Concatenated Sequences)	Nucleotide Average Identity * with Prototypes from the Same Clade
VZV001	Female	48	Not detected	WT	1	1203 bp	99.2%
VZV002	Female	72	Not detected	ND	NS	-	-
VZV003	Male	67	Not detected	ND	NS	-	-
VZV004	Female	58	HAdv	WT	2	909 bp	99.6%
VZV005	Male	67	Not detected	ND	NS	-	-
VZV006	Male	48	EBV	WT	NS	-	-
VZV007	Male	79	Not detected	WT	NS	-	-
VZV008	Male	17	Not detected	WT	NS	-	-
VZV009	Male	53	Not detected	WT	NS	-	-
VZV010	Male	32	Not detected	ND	NS	-	-
VZV011	Female	31	Not detected	ND	NS	-	-
VZV012	Female	34	EBV	WT	5	804 bp	99.7%
VZV013	Female	57	Not detected	WT	NS	-	-
VZV014	Female	62	Not detected	WT	NS	-	-
VZV015	Male	38	Not detected	WT	1	507 bp	100%
VZV016	Female	51	Not detected	WT	2	1203 bp	99.45%
VZV017	Female	30	Not detected	WT	NS	-	-
VZV018	Male	28	Not detected	WT	6	507 bp	100%
VZV019	Male	30	Not detected	WT	NS	-	-
VZV020	Male	41	Not detected	WT	1	213 bp	100%
VZV021	Female	17	Not detected	WT	3	906 bp	99.9%
VZV022	Male	35	Not detected	WT	NS	-	-
VZV023	Female	31	Not detected	WT	1	804 bp	99.8%
VZV024	Female	72	Not detected	WT	NS	-	-
VZV025	Female	66	Not detected	WT	NS	-	-
VZV026	Male	42	Not detected	WT	1	696 bp	-
VZV027	Male	39	Not detected	WT	5	1203 bp	99.5%
VZV028	Female	20	Not detected	WT	NS	-	-
VZV029	Female	30	Not detected	WT	NS	-	-
VZV030	Male	28	Not detected	WT	5	906 bp	99.2%

EBV, Epstein–Barr virus; HAdv, Human Adenovirus; WT, wild-type; ND, not discriminated; NS, not sequenced; bp, base pair; * nucleotide average identity calculated with Ezbiocloud online tool [33].

**Table 3 viruses-17-00286-t003:** Details of VZV isolates in relation to their genetic classification (Clade), year of CSF collection, and respective GenBank accession number.

Sample	VZV Clade	CSF Collection Year	ORF 22 Accession Numbers	ORF 38 Accession Numbers	ORF 54 Accession Numbers	ORF 62 Accession Numbers
VZV001	1	2018	OR770642	OR885927	OR992036	OR988147
VZV004	2	2018	OR958829	NS	OR992037	OR988148
VZV012	5	2019	OR988144	OR898439	OR992038	NS
VZV015	1	2019	NS	OR898440	OR992039	NS
VZV016	2	2019	OR988145	OR898441	OR992040	OR988149
VZV018	6	2019	NS	OR898442	OR992041	NS
VZV020	1	2018	NS	NS	PP003316	NS
VZV021	3	2019	NS	OR898443	PP003317	OR988150
VZV023	1	2019	OR988146	OR898444	PP003318	NS
VZV026	1	2019	NS	OR915870	PP003319	NS
VZV027	5	2019	OR689713	OR958827	PP003320	OR988151
VZV030	5	2019	NS	OR958828	PP003321	OR988152

NS, ORF not sequenced. Note: ORF 22: nucleotides 37,837 to 38,283; ORF 38 nucleotides 69,250 to 69,599; ORF 54 nucleotides 95,109 to 95,330; ORF 62 nucleotides 106,983 to 107,401; nucleotide positions were based on the reference sequence X04370.1.

## Data Availability

All available data regarding this manuscript is presented in the present manuscript.

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
