# Peer review of "Phylogenetic Analysis of Varicella–Zoster Virus in Cerebrospinal Fluid from Individuals with Acute Central Nervous System Infection: An Exploratory Study"

_viruses, 2025, doi:10.3390/v17020286_

Round 1
Reviewer 1 Report
Comments and Suggestions for Authors
This manuscript analyzed 30 varicella-zoster virus (VZV)-positive samples identified from a cohort of 600 patients with acute central nervous system infection, providing valuable insights into VZV in Brazil. Notably, it identified previously unreported VZV clades (Clades 2 and 6) in the region. However, there are areas that could be improved:
1. Among the 600 samples, only 30 were VZV-positive, and just 12 of these underwent genetic characterization. This limited sample size restricts the generalizability of the findings.
2. The study would benefit from additional analyses, such as detailed statistical comparisons, assessments of nucleotide diversity, and sequence alignments to evaluate genetic differentiation across clades identified in the cerebrospinal fluid samples. These enhancements would strengthen the discussion and provide deeper insights into VZV evolution and genetic variability in Brazil.
Author Response
Comment 1: Among the 600 samples, only 30 were VZV-positive, and just 12 of these underwent genetic characterization. This limited sample size restricts the generalizability of the findings.
Response 1: We certainly agree with this important observation. We have modified the text to include this clarifying information. We now point out that the low and variable concentration of VZV in individual cerebrospinal fluids as well as the fact that a portion of our subjects were already undergoing anti-viral treatment prior to sample collection are limitations of our investigation. Nevertheless, we strongly believe that our investigation is novel and of considerable value. Please check the Discussion section, lines 352-354. Also, we have modified the Title of the manuscript to emphasize these limitations, as follows: “Phylogenetic Analysis of varicella-zoster virus in cerebrospinal fluid from individuals with acute central nervous system infection: an exploratory study”.
Comment 2: The study would benefit from additional analyses, such as detailed statistical comparisons, assessments of nucleotide diversity, and sequence alignments to evaluate genetic differentiation across clades identified in the cerebrospinal fluid samples. These enhancements would strengthen the discussion and provide deeper insights into VZV evolution and genetic variability in Brazil.
Response 1: We appreciate these thoughtful comments from the reviewer. To detail statistical analysis on diversity, we previously conducted an analysis using MEGA v11 software. This software provides a Disparity Index test to assess the homogeneity of substitution patterns (with p values below 0.05) to determine whether sequences have evolved under different nucleotide substitution processes, as described by Kumar and Gadagkar (DOI: 10.1093/genetics/158.3.1321). We decided not to include the results of this analysis in the manuscript because the sequences obtained by VZV genotyping are commonly short in relation to the GenBank prototypes. These MEGA findings did not include any additional information to the patterns observed in the phylogenetic tree as well as the mean genetic distance calculations that are already presented in our results. We believe this data collectively is sufficient to demonstrate the high genetic stability of the VZV genome. Genetic data interpretation is already discussed through the phylogenetic analysis of these partial VZV DNA sequences.

Reviewer 2 Report
Comments and Suggestions for Authors
Paiao et al., studied the CSF of 600 hospitalized Brazilian patients hospitalized with central nervous symptoms. From 5% of the persons VZV were detected. Isolates of clades 2 and 6 have never been detected in Brazil before. Their study provides data about the present VZV epidemiological situation in Brazil and possible connections of their isolates to VZV strains of other continents.
Abstract.
No abbreviations in the abstract chapter.
line 30-31 - The sentence has no meaning. Molecular characterization ……
VZV – somewhere the official name „Human herpesvirus 3” and the common name Chickenpox should also be mentioned.
In the introduction chapter besides genetic data some sentences should be given about the symtoms, VZV epidemiological data of the present and past, its story in Brazil, about the disease itself.
The authors should indicate here, that detection of VZV in CSF does mot indicate that virus is causative agent of the symptoms, more probably they found sign of reactivation of a latent herpesvirus.
How were these causes diagnosed in the hospitals?
Serum samples were not available from the hospitals? To compare CSF and serum data would be beneficial.
Mat and Meth.
line 96 – cytomegalovirus is not a pathogen.
2.2. If the authors studied 25 more agents why are not some data of them given?
2.5. The developed PCR product was not too small to be adequately identifiable in gel and to be sequenced? If the authors used the inner primer for sequencing no readable sequence remained, as 20-30 nt downstream of the sequencing primer is not readable. How long readable sequence the authors got from a single sample?
A 30 +-ból mi alapján választották ki a 12 szekvenálandót?
Results:
3.1.1. - And children?
line - 168 - Why 12? Why were these 12 samples selected for sequencing?
line 172 - In the text two EBV coinfections are mentioned, in Table 1 only one.
line 182 – reason?
line 190 – Why was not 100% for all?
Discussion
Were the patients with + VZV diagnosis under any immunsuppressive effect (therapeutic drugs, or HIV?).
spelling:
line 48, 49, 52, 54 etc. reported – space - [12]. A space is necessary between the word and the bracket.
line 65, 83 – point is not in good position
Table 1. size of the numbers/letter 62r is not correct
Author Response
Comment 1: No abbreviations in the abstract chapter.
Response 1: We thank the reviewer for this observation. Please check the Abstract. We have removed all abbreviations from it.
Comment 2: Line 30-31 – The sentence has no meaning. Molecular characterization ……
Response 2: We thank the reviewer for this observation. We have rephrased the sentence. Please check lines 30 to 32.
Comment 3: VZV – somewhere the official name „Human herpesvirus 3” and the common name Chickenpox should also be mentioned.
Response 3: We have added this information. Please observe in the Introduction lines 49 to 51.
Comment 4: In the introduction chapter besides genetic data some sentences should be given about the symptoms, VZV epidemiological data of the present and past, its story in Brazil, about the disease itself.
Response 4: We appreciate your suggestion. Regarding clinical data on the cases discussed in the present study, we have included the information in Material and Methods, lines 84-86. Regarding data on VZV epidemiological data of the present and past, its history in Brazil, as well as data on VZV vaccination in the country we have added this information in Discussion, lines 309 to 316.
Comment 5: The authors should indicate here that detection of VZV in CSF does not indicate that virus is causative agent of the symptoms, more probably they found sign of reactivation of a latent herpesvirus.
Response 5: We thank the reviewer for raising this point. We certainly agree that it is necessary to be cautious when detecting herpes virus in CNS, since these viruses can establish latency, and their detection may just be an incidental finding. However, regarding the role of VZV as the causative agent of symptoms, in the present study, we respectfully disagree with the reviewer. In the present series of cases, according to medical records, VZV was interpreted as the causative agents of the symptoms and patients were managed as so, receiving specific antiviral therapy with acyclovir. In fact, in other cases, as mentioned in references 3 to 8, involving similar situations, VZV has been associated with acute neurologic syndromes, as meningitis and encephalitis, as presently described in our study.
Comment 6: How were these causes diagnosed in the hospitals?
Response 6: As mentioned in Material and Methods, VZV was diagnosed at the hospitals, where the patients were first attended. Please observe lines 104-116.
In summary: All CSF samples underwent a complete cell count, differential leucocyte count, and biochemistry tests (including glucose, protein, and lactate concentrations) to analyze CSF parameters. For pathogen analysis, the following tests were performed: examination of Gram-stained and India Ink smears; bacterial and fungal cultures; and use of the BioFire® FilmArray Meningitis/Encephalitis Panel (BioMérieux) and XGEN Multiplex Neuro 9 (Mobius Life).
Comment 7: Serum samples were not available from the hospitals. To compare CSF and serum data would be beneficial.
Response 7: We appreciate the thoughtful observation. However, serum samples were not available to be included in the analysis.
Mat and Meth.
Comment 8: Line 96 – cytomegalovirus is not a pathogen.
Response 8: We certainly agree with this observation and appreciate the observation. However, we do not fully understand the specific concern being raised. Cytomegalovirus is one of the opportunistic pathogens diagnosed by FilmArray and XGEN Multiplex Neuro 9 platforms, as described in the manufacturer’s instructions. In our study, the detected co-pathogens were not associated with the acute infection. As mentioned above we understand that the detection of herpes viruses in CNS may just be an incidental finding, due to their latency in CNS.
Comment 9: If the authors studied 25 more agents why are not some data of them given?
Response 9: We thank the reviewer for this comment. However, as stated in the abstract, the purpose of the present study was specifically the molecular characterization of Varicella-Zoster virus isolates in cerebrospinal fluid from individuals presenting acute central nervous system infection. Please check lines 30-32. Characterization of other agents in CSF was beyond the scope of the present investigation.
Comment 10: 2.5. Was the developed PCR product not too small to be adequately identifiable in gel and to be sequenced? If the authors used the inner primer for sequencing no readable sequence remained, as 20-30 nt downstream of the sequencing primer is not readable. How long readable sequence the authors got from a single sample?
Comment 10: We thank the reviewer for raising this point. To address this point, we chose to amplify specific ORFs, as proposed in the literature. The concatenated nucleotide fragments of 1205 pb were obtained after amplification of selected ORfs commonly used for genotyping studies. To clarify this, we have added a sentence. Please check lines 205 to 207.
Comment 11: A 30 +-ból mi alapján választották ki a 12 szekvenálandót?
Response 11: We do apologize but the question was not clear. In fact, we did not understand the question. It does not seem to be written in English.
Results:
Comment 12: 3.1.1. – And children?
Response 12: -We thank the reviewer for raising this point. In fact, there were no children included in this series of VZV-positive cases. Patients’ age ranged from 17 to 79, as presented in Results.
We did not identify VZV cases in children likely because Brazilian children routinely receive the v-Oka vaccine. As discussed in our manuscript (lines 313-316) : vaccination for VZV is associated in the reduction of the number of severe varicella cases reported in many countries [42], as well as a decreasing in the number of severe varicella cases among children under 0 to 5 year old age group [43].
Comment 13: Line – 168 – Why 12? Why were these 12 samples selected for sequencing?
Response 13: We agree with the reviewer the explanation for that aspect of the result was not clear in the previous text. In fact, out of 30 VZV samples (identified by PCR) only twelve samples were successfully amplified for ORFs 22, 38, 54, and/or ORF 62, enabling VZV Clade classification. We have rephrased the sentence to make it clearer. Please check on lines 203-204.
Comment 14: Line 172 – In the text two EBV coinfections are mentioned, in Table 1 only one.
Response 14: We thank the reviewer for this comment. We have rewritten the text to make it clear. EBV was detected using FilmArray and XGEN Multiplex Neuro 9 commercial platforms. Please check lines 188-189, and Table 2.
Comment 15: Line 182 – reason?
Response 15: Please check the answer above.
Comment 16: Line 190 – Why was not 100% for all?
Response 16: Our interpretation is that genetic variation in VZV region of the ORFs might hinder primer binding to these viral genomes. To address this point, we chose to amplify specific ORFs, as proposed in the literature. Semi-nested PCR was performed to obtain DNA fragments of four ORFs of the VZV genome (ORF 22, ORF 38, ORF 54 and ORF 62), using primers and cycling parameters as previously described [11,19,20,28,29]. Please check lines 134-137.
Discussion
Comment 17: Were the patients with + VZV diagnosis under any immunsuppressive effect (therapeutic drugs, or HIV?).
Response 17: We were unfortunately unable to obtain complete clinical data on all of the VZV-positive patients, and now state this as a study limitation in Discussion.
Spelling:
Comment 18: Line 48, 49, 52, 54 etc. reported – space – [12]. A space is necessary between the word and the bracket.
Response 18: We thank the reviewer for the careful analysis of the manuscript. We corrected all the spaces as suggested.
Comment 19: Line 65, 83 – point is not in good position
Response 19: We thank the reviewer for the careful analysis of the manuscript. We corrected all points as suggested.
Comment 20: Table 1. size of the numbers/letter 62r is not correct
Response 20: We thank the reviewer for the careful analysis of the manuscript. We corrected the text of the manuscript, as suggested.

Round 2
Reviewer 1 Report
Comments and Suggestions for Authors
The authors have effectively addressed the key concerns raised in the initial review. However, there are a few minor concerns:
- Response to Original Comment 2: While the rationale for excluding the Disparity Index test results is justified, including a supplementary table summarizing these findings could enhance completeness.
- The manuscript now clarifies that Clades 2 and 6 were identified in Brazil for the first time. If possible, a brief discussion comparing the presence of these clades with global VZV phylogenetic distributions would be valuable, particularly in assessing whether their occurrence in Brazil suggests regional transmission patterns or introductions from other geographical regions.
Author Response
Comment 1- Response to Original Comment 2: While the rationale for excluding the Disparity Index test results is justified, including a supplementary table summarizing these findings could enhance completeness.
Response 1: We appreciate the reviewer's thoughtful comment. As requested, we have included the Disparity Index analysis as a supplementary table. This additional table is now cited in lines 303–306. “Other than the stability of the VZV genome, demonstrated in molecular analysis, additional data that may further enhance the consistency of our findings is provided in the Supplementary Material (see Table S2 – Test of the Homogeneity of Substitution Patterns between Sequences).”
Comment 2 - The manuscript now clarifies that Clades 2 and 6 were identified in Brazil for the first time. If possible, a brief discussion comparing the presence of these clades with global VZV phylogenetic distributions would be valuable, particularly in assessing whether their occurrence in Brazil suggests regional transmission patterns or introductions from other geographical regions.
Response 2: We appreciate the thoughtful comment from the reviewer. As requested, we included a brief discussion in lines 350 to 359 to complete the discussion, stating where VZV clades are pronounced predominant and that our results not necessarily identify the introduction of these clades in Brazil and new references (51, 52, 53).
“Although the global distribution of VZV is well defined [15,45], its genetic diversity has been insufficiently investigated to clarify its differential clinical impact and its potential epidemiological implications.
VZV clades may exhibit geographic distinctions [50], with certain clades showing a pronounced predominance in specific regions. Clades 1, 3, and 5 are more prevalent in America [9], Africa [10], Europe [51], and Oceania [11], while Clade 2 is more commonly found in Asia [49,52]. Clade 6 has been described in only a few instances, as it was newly proposed after 2008 [53]. The previously undetected circulation of Clades 2 and 6 may be attributed to limited sampling in earlier studies — or even the absence of studies — rather than a recent introduction of these clades in Brazil.”
